# Intervention Programme Based on Self-Determination Theory to Promote Extracurricular Physical Activity through Physical Education in Primary School: A Study Protocol

**DOI:** 10.3390/children10030504

**Published:** 2023-03-03

**Authors:** Rubén Llanos-Muñoz, Mikel Vaquero-Solís, Miguel Ángel López-Gajardo, Pedro Antonio Sánchez-Miguel, Miguel Ángel Tapia-Serrano, Francisco Miguel Leo

**Affiliations:** 1Grupo Análisis Comportamental de la Actividad Física y el Deporte (ACAFYDE), Departament of Didactics of Musical, Plastic and Corporal Expression, Teaching Training College, University of Extremadura, 10003 Cáceres, Spain; 2Grupo Análisis Comportamental de la Actividad Física y el Deporte (ACAFYDE), Departament of Didactics of Musical, Plastic and Corporal Expression, Faculty of Sports Sciences, University of Extremadura, 10003 Cáceres, Spain

**Keywords:** motivation, students, experimental study, interpersonal style, need support, sport

## Abstract

Low levels of physical activity (PA) are a concern among students, producing negative physical, health and mental consequences. This study aims to present a protocol intervention in physical education (PE) based on self-determination theory (SDT) to enhance students’ motivation towards performing PA and increasing their PA levels in their leisure time. This protocol is a convenience study with two allocation arms (intervention group versus control group). SDT-based motivational strategies will be carried out and co-created with PE teachers to increase motivation and out-of-school PA levels. Data collection will be conducted three times: before the intervention, after the intervention (four months after baseline) and at the end of the intervention (retention measurement, seven months after baseline). The measures will assess perceived teacher support for PA, motivation towards PA, intention to be physically active, PA levels, engagement in PE and academic performance. Overall, this intervention programme is expected to increase students’ autonomous motivation for PA and their PA levels in their free time. This intervention might encourage teachers to establish strategies and resources to increase their students’ adaptive outcomes.

## 1. Introduction

Regular physical activity (PA) has been associated with many physical, psychological, social, and cognitive benefits [1,2,3,4]. To achieve these benefits, the World Health Organization recommends that children and adolescents aged 5–17 years engage in at least 60 min of moderate-to-vigorous intensity PA (MVPA) per day [1]. Despite these benefits, the latest report on global PA trends, which included 1.6 million students worldwide from 146 different countries, found that more than 81% of 11–17-year-olds do not meet the daily PA recommendations [5]. Likewise, the Global Matrix 4.0 report for PA in 57 countries highlighted that, at the European level, only 34–39% met the PA recommendations [6].

Since school is the place where students expend most of their time, this context is crucial to explain and encourage specific behaviours (e.g., cognitive and academic development, development at behavioural, cellular, functional and morphological levels) [4]. Although the educational context is a favourable environment to generate healthy behaviours such as PA, the fact is that boys and girls spend a lot of time seated [7]. It has been shown that there is a need to develop strategies to encourage PA practice in students as PE levels during school hours are not enough [7]. In this sense, motivation is a fundamental variable to understand, explain, and improve positive consequences in and out of the classroom. Nevertheless, to identify the motivational parameters of students’ motivation for PA, we must determine the reasons for practising PA [8]. Previous research has shown that school interventions based on behaviour change theories (e.g., self-determination theory [SDT], goal perspectives theory, social cognitive theory) are more effective in promoting PA [9]. Therefore, considering the few existing studies that have promoted in-school and out-of-school PA, more intervention studies are needed to promote motivation and adherence towards PA and alleviate dropout from PA over time [7].

### 1.1. Physical Activity and Self-Determination Theory

Motivation has emerged as a crucial antecedent of students’ development and learning in the school context [10]. For this reason, SDT [8,11] has been widely used to promote motivation for PA and PA levels in students. As defined in SDT, motivation fluctuates seamlessly between higher and lower levels of self-determination, which refers to the extent to which behaviours are voluntary or self-determined [12,13]. In this sense, motivational behaviour can be fragmented into three types of motivation [14]: autonomous motivation, controlled motivation and amotivation, each determined by different forms of motivational regulation ranging from the highest to the lowest level of regulation. In autonomous motivation, behaviours are given by integrated regulation (the activity is assimilated within the self), identified regulation (achieve an action in line with the person’s values) and intrinsic regulation (performing an activity for its enjoyment or because it is challenging). In controlled motivation, behaviours are assessed by external (execute an action to get a reward or avert chastisement) and introjected (performing an activity to feel good and avoid feelings of guilt or anxiety) regulation. Finally, at the opposite extreme of motivation is amotivation, which is the absence and lack of interest and motivation.

SDT defends the existence of three basic psychological needs (BPN) that can be supported or thwarted by the social environment: autonomy, competence, and relatedness needs [15]. In the educational context, the physical education (PE) teacher can be crucial in supporting or thwarting these three needs [16,17,18]. The teaching behaviours concerning out-of-school activities during PE classes can encourage their students’ practice of extracurricular PA, supporting their autonomy, competence, and relatedness [19,20]. Specifically, PE teachers’ autonomy support to extracurricular PA implies using strategies to encourage students’ initiative, and provide choices, options, and opportunities to do active sports and/or exercise in their free time. Concerning extracurricular PA, competence-supportive PE teachers offer their students challenging extracurricular activities, show confidence in their capacity to effectively engage in these types of PA, express clear expectations and provide activities adjusted to the students’ skill levels, helping them to achieve their goals and giving positive feedback when they perform active sports and/or exercise in their free time. Finally, relatedness-supportive PE teachers express affection to their students when they share their experiences about their leisure-time PA, they propose out-of-school activities with peers, and they dedicate time and energy to listening to their students and helping them feel socially connected [15,18,19,20]. In this sense, PA and fitness are related to different types of changes in the brain relevant to cognitive function and learning [21,22,23,24].

Previous studies have shown how a need-supportive interpersonal style is a predictor of a more self-determined motivation, which in turn, can predict the intention to be physically active [16], knowledge [17], engagement [18], and performance in PE [18]. Furthermore, it is essential to understand that the activities teachers develop in their classrooms can affect students’ extracurricular lives. For example, authors such as Tilga et al. [25], Tilga et al. [26] and Hagger et al. [19] have found that students who perceive that their PE teacher supports autonomy show higher levels of autonomous motivation for PA and greater intention to perform PA in their free time. Therefore, it can be concluded that extracurricular PA-supportive PE teachers can impact students’ out-of-school behaviours, positively impacting the quality of motivation and intentions to engage in PA and PA levels [27]. Based on previous studies, one of the main limitations of school-based interventions is their low sustainability once the research team stops implementing the intervention. Although there are several reasons that may cause this abandonment of implementation, lack of teacher training is one of the main reasons. Therefore, previous literature suggests that training and involving teachers in the intervention programme could positively contribute to the sustainability of the school-based intervention [28].

### 1.2. Previous Intervention Studies Based on SDT in the School Context

Previous systematic reviews and meta-analyses on SDT-based school interventions have shown that teachers who support students’ motivational processes generate adaptive consequences towards PA [29,30], but the effect of these interventions is low [8]. In this sense, previous studies have implemented motivational strategies or used a multicomponent theoretical design (i.e., intervening from different perspectives, areas, or actors), combining different approaches with SDT to obtain positive effects on PA behaviour [8,31,32,33,34,35]. Concerning SDT, Kelso collected thirty studies using this theory as a theoretical framework [8]. However, analysing these researches in detail, only two studies—Moreno-Murcia and Sánchez-Latorre [36] and Kokkonen et al. [32]—developed a training programme with PE teachers, measuring the out-of-school outcomes; that is, students’ motivation for leisure-time PA and their out-of-school PA levels. Moreover, only Moreno-Murcia and Sánchez-Latorre [36] found significant differences in intrinsic motivation and PA levels after the intervention. Thus, the results should be interpreted with caution. Given that the evidence of out-of-school programmes is limited, more research is needed to identify the most appropriate strategies to elicit more self-determined forms of motivation and increase students’ out-of-school levels of PA.

### 1.3. The Present Study

In accordance with the previous studies and the theoretical framework of SDT, we highlighted that: (a) most of the studies were based on different theories; (b) they assessed only variables within PE; (c) they were applied to young people over 12 years of age; (d) PE teachers were not always involved in the intervention; and (e) most of the strategies developed only focused on autonomy support. Therefore, this study aims for the PE teacher to apply a series of SDT-based motivational strategies that support autonomy, competence, and relatedness towards PA in primary education students to improve their motivation for PA and, consequently, their extracurricular PA. For this purpose, an intervention will be carried out in students (10 to 12 years), where strategies will be applied to support the three BPNs (i.e., autonomy, competence, and relatedness) concerning PA and with the teachers playing a more active role, as they will co-create and develop these motivational strategies to promote motivation and out-of-the-school practice of PA.

## 2. Methods

### 2.1. Design and Participants

This quasi-experimental study will be carried out in Cáceres (Extremadura, Spain). Convenience sampling will be conducted in six different Primary Schools. Three schools will be randomly assigned to the control school and the other three schools to the experimental school. The sample size will be estimated using the following formula: *n* = (Z)^2^ (*p*(1 − *p*) e^2^), where “*n*” is the sample size, Z = 1.96 (95% confidence interval), *p* = number of 4th and 5th primary school students in the city where the study will be conducted (±3350 students), and e = margin of error (3%). The minimum sample size (considering a 10% non-response) will be 345 students. All the schools will have similar socio-demographic and built-environment characteristics [37]. School courses in Extremadura run from early September to mid-June, with a Christmas break from late December to early January and a break around March or April for the Easter holidays. As seen in Figure 1, assessments will be completed at baseline at the beginning of the school course (Term 1, February) and after the intervention (Term 2, June). A follow-up measurement will be conducted at the beginning of the following school course (Term 3, October). The study has been agreed upon by the Ethics Committee of the University of Extremadura (120/2018) and follows the guidelines of the Declaration of Helsinki.

#### 2.1.1. Selection Process and School Characteristics

A roster of the primary schools in the city of Cáceres will be drawn up. This list will be categorised according to the nature (public or private) of the potential participating schools, resulting in 26 schools. Subsequently, an explanatory letter with the project’s objectives will be sent to the principals, who will be asked to reply to the e-mail if they decide to participate to receive a more detailed explanation of the project. Schools will have a maximum of 15 days to confirm their participation. As the project will be conducted in PE lessons, the PE teachers must support the principals.

#### 2.1.2. Teachers’ Training Program

PE teachers from participating schools will be involved in a training programme on motivational strategies. The research team will contact teachers who decide to participate in the project by e-mail or mobile phone. This contact aims to guarantee their participation in the project and present the objectives and characteristics of the intervention programme. Teachers who join the PE staff after the intervention programme should establish contact to ensure the full development of the programme. The research team will provide each teacher with a document about the objectives to be achieved in the project and the methodology to be implemented.

Regarding the training programme, at least two sessions will be held in February with the teachers whose schools were designated as the experimental school, each lasting 4 h. In the first session, the explanation and timing of the project and its theoretical basis will be presented. Firstly, the research team will explain the fundamentals of SDT applied to PE, underscoring the promotion of PA practice based on supporting the BPNs. Secondly, the teachers will be divided into work groups for a brainstorming session to co-create different intervention strategies where this theoretical basis can be put into practice to promote autonomy, competence, and relatedness towards PA. These strategies will be shared with the rest of the teachers and supervised by the research team. During the second training session, the research team and the teachers will discuss these strategies’ feasibility and functionality to promote PA both in educational and extracurricular contexts. In other words, the ideas developed in the brainstorming session can be adapted correctly and appropriately to the schools’ educational reality. In addition, the research team will arrange weekly meetings with the teachers to obtain feedback on the strategies developed.

#### 2.1.3. Students’ Characteristics

All 4th- and 5th-graders from the participating schools will be invited to be involved. The parents’ or legal guardians’ consent and the students’ agreement will be required to participate in the study. Finally, the inclusion criteria for participation in the intervention programme will be: (i) to have the families’ or legal guardians’ and students’ consent to participate; (ii) to complete the questionnaires of the study variables; and (iii) not to participate in another programme with similar characteristics during the intervention. The selection of these ages is based on the fact that studies confirm that the adolescent stage is when a number of sport dropout behaviours occur and there are higher rates of inactivity [38,39]. Hence the importance of generating good habits from an early age.

### 2.2. Intervention Based on SDT in Primary School

In February, an informative document will also be given to the families or legal guardians, accompanied by the informed consent to be signed. The research team will go to the schools to collect the data on certain days according to the organisational availability of the teachers during lessons. There will always be at least one research team member in the classroom while the students complete the questionnaires to solve any possible doubts. The questionnaires will be completed on paper in approximately 40 min. The variables described in Section 2.3 will be measured three times: (i) before the intervention (February); (ii) after the intervention (June); and (iii) a follow-up measure at the beginning of the following school year under the same conditions as in the two previous measures (October).

Given that there is no set duration as the benchmark for an intervention programme to be effective, it was decided to set a duration that was in line with previous programmes carried out in primary education [36,40,41,42]. The intervention programme for promoting motivation and PA in primary schools will be carried out over approximately 4 months (March to June). During this period, the strategies that were co-created jointly with the PE teachers in the schools that form part of the experimental group will be developed. Approximately 10 strategies are expected to be developed to involve different agents (teachers, families, and peers) both in the school and in the out-of-school context. For instance, some possible strategies are introducing new physical-sports activities or sports modalities during free time, sharing the PA carried out outside the school context with the class, involving the families, organising complementary and extracurricular activities that encourage the practice of PA, etc. In this sense, each strategy will be associated with promoting a specific BPN, assuming that each one can favour at least one BPN. The reason for designing these strategies is to ensure that the students acquire a level of autonomy that determines when, how and why to do PA and adequate competence to manage the surrounding environment, face new challenges and acquire specific skills. Moreover, this sporting practice can be performed in the company of friends and family to generate a broader social bond, using PA as a nexus of union between people who intend to practice physical sports activities with other people for greater enjoyment. Each experimental school will carry out the same strategies. However, each teacher will be able to apply them as they see fit, adapting them to their own educational reality, under the supervision of a member of the research team (see Figure 2).

In this regard, we highlight the role of the facilitator within the research team. The supervisor or facilitator role has been used in many contexts to encourage the development of behaviours that significantly benefit individuals [43,44,45]. For a better understanding, the definition established by the Royal Spanish Academy of the facilitator is: “A person who acts as an instructor or guide in an activity”. Facilitators will fulfil various functions to ensure the compliance and correct development of the programme, as well as to assist when necessary in carrying out strategies that are not adequately developed and/or are difficult to implement in that educational context. The main functions of the facilitators will be: (a) to support the teaching team in explaining and developing the strategies in the educational context; (b) to provide all the (physical) material to the different educational centres participating in the study; (c) to carry out weekly monitoring of the fulfilment of the strategies by the teachers; (d) to design a programme of physical-sports activities that include different games and/or sports modalities for break time; and (e) to design a programme of activities for complementary and extracurricular activities to be carried out during the project.

According to objective (c), this monitoring will be carried out through checklists in which items related to teachers’ autonomy support will be collected (see Appendix A). The lists should be completed periodically to obtain an adequate follow-up of the intervention and, thus, to help teachers to develop some of the more complex strategies. In the Appendix A, a checklist is provided with general items based on the PASSES [19], to encourage students’ autonomy with support from teachers. This checklist will be complemented with items that address the strategies co-created with the PE teachers in the training programme so they can be adjusted to the school’s educational reality.

### 2.3. Measures

The following measures were applied to evaluate the variables at three different times (February, June, and October) in the study participants to determine and identify which students could be part of the intervention programme based on psychosocial and physiological dimensions. To summarise, all the instruments are shown in Table 1.

*Socio-economic Status.* The FAS Scale [46] is an index composed of four items that are indicators of material wealth: How many cars or vans does your family own (0, 1, 2 or more)? Do you have a room to yourself (0, 1)? During the last 12 months, how many times did you go on holiday with your family (0, 1, 2, 3 or more)? How many computers does your family own (0, 1, 2, 3 or more)? The items are scored from 0 to 10. The higher the score, the higher the family’s socioeconomic status. 

*Height and Weight.* Students’ height and weight will be assessed by trained research staff of the same sex as the student. Weight will be measured with a digital electronic scale (model SECA 877), and height will be assessed with a telescopic height-measuring instrument. Both measurements will be taken twice, and the mean value will be used to calculate the body mass index.

*Perceived Need Support in PA.* The Spanish translation of the Perceived Autonomy Support Scale for Exercise Settings [19], adapted by Moreno-Murcia et al. [47], will be used. This scale has 12 items (i.e., “I feel that the PE teacher gives me choices and opportunities about whether to do sport and/or vigorous exercise in my free time”) grouped into a single factor (teacher autonomy support). The response format is a Likert scale ranging from 1 (strongly disagree) to 7 (strongly agree).

*Motivation for Physical Activity.* To assess motivation to practice physical-sports activity, an adaptation of the Behavioural Regulation in Exercise Questionnaire (BREQ-3; [48]), validated for the Spanish context by González-Cutre et al. [49], will be used. This scale begins with stem “I exercise because…”, followed by 23 items, grouped into six factors that measure intrinsic motivation (i.e., “it’s fun and satisfying to exercise”), integrated regulation (i.e., “it is consistent with my lifegoals”), identified regulation (i.e., “I think it’s important to make the effort to exercise regularly”), introjected regulation (i.e., “I feel a failure when I haven’t exercised in a while”), external regulation (i.e., “I feel under pressure from friends/family to exercise”), and amotivation (i.e., “I think exercise is a waste of time”). All the factors have four items except for the identified regulation, which has three. The response format is a Likert scale ranging from 1 (totally disagree) to 5 (totally agree).

*Intention to be Physically Active.* We used the Intention to be Physically Active (MIFA) questionnaire, created by Hein et al. [50] and adapted to the Spanish context by Moreno-Murcia et al. [51]. This instrument was developed with Spanish primary school students [52]. The scale is preceded by the phrase, “Regarding your intention to take up a sporting activity ...”, and comprises 5 items (i.e., “I am interested in developing my physical fitness”). The response format is a Likert scale ranging from 1 (totally disagree) to 5 (totally agree).

*Perception of Physical Activity Levels.* The Spanish-validated scale of Benítez-Porres et al. [53] was used to assess PA in children. This self-administered questionnaire measures moderate to vigorous PA performed in the last seven days in students. It consists of ten items, nine of which are used to calculate the activity level. The other item assesses whether any illness or other event prevented the child from doing their regular activities in the last week. The items are scored from 1 to 5.

*Engagement in Physical Education.* To assess students’ engagement in PE, we will use Skinner et al.’s [54] Student Engagement Questionnaire, adapted by Shen et al. [55]. The scale is preceded by the stem “When I am in PE class…”, and comprises 5 items (i.e., “I work as hard as I can”), which address students’ perceptions of their endeavour, heed, and steadfastness in PE classes. The response format is a Likert scale ranging from 1 (totally disagree) to 5 (totally agree).

*Academic Performance.* Reference to the grades obtained in the 2nd and 3rd trimesters in the subjects of PE, first language (Spanish), mathematics and second language (English). The grade point average (GPA) will then be calculated as an average of the scores in these four subjects. Previous studies have used these subjects as indicators to assess academic performance [56,57].

In addition, to evaluate the process and development of the school-based intervention, semi-structured interviews will be carried out with teachers, students and families in order to obtain complementary information to that obtained from the different questionnaires and scales mentioned above.

**Table 1 children-10-00504-t001:** Summary variables and instruments to measure them used in study.

Variable	Instrument
Socio-economic status	FAS Scale [46]
Height	Stadiometer
Weight	Weighing machine
Perceived Need Support in PA	PASSES [47]
Motivation toward physical activity	BREQ-3 [49]
Intention to be physically active	MIFA [51]
Perception of physical activity levels	PAQ-C [53]
Engagement in physical education	Physical Education Engagement Questionnaire [55]
Academic performance	Grades obtained in 1st and 2nd trimester

### 2.4. Statistical Analysis

The primary outcome will be changes in the levels of students’ motivation for out-of-school PA. Secondarily, perceived exercise autonomy support, the intention to be physically active, perception of PA levels, engagement in PE and academic performance will also be assessed. For this purpose, firstly, the nature of the variables will be taken into account through the application of normality, validity, and reliability tests. Secondly, the principles of independence, normality, linearity and variance of the study variables will be tested for general linear model analyses. Thirdly, intergroup and intragroup differences will be analysed with repeated-measures ANOVA or nonparametric tests such as McNemar’s or the Kruskall–Wallis tests, as appropriate. For continuous variables, Bonferroni-corrected multiple paired *t*-tests will be calculated to determine intra-group (i.e., between the experimental and control group) and inter-group (i.e., between the pre-test and post-test) differences. For categorical variables, a chi-square test will be performed. While Cramer’s V will be used to describe the degree of association between categorical variables and experimental and control schools, McNemar’s test will be used to analyse baseline and post-intervention differences in categorical variables in experimental and control schools, respectively. Effect sizes will be assessed using Partial Eta Squared (η_p_^2^) and Cramer’s V values for continuous and categorical variables, respectively. Effect sizes will be considered small, moderate or large when η_p_^2^ is greater than 0.01, 0.06 and 0.14, respectively, and when Cramer’s V is greater than 0.10, 0.30 and 0.50, respectively. Significance will be set at *p* < 0.05 for all analyses.

## 3. Discussion

PE can be considered a potential tool to develop interventions aimed at promoting PA in students, in line with the curricular objectives addressed in this subject [58]. However, the number of studies examining interventions to promote school PA is relatively small, and more interventions should be reviewed and analysed in more detail. The goal of the study of this protocol is for the PE teacher to apply a series of SDT-based motivational strategies for PA that support autonomy, competence, and relatedness in primary education students to improve their motivation for in-class PA and, consequently, for extracurricular PA. The intervention will be grounded in the SDT [13] and is expected to make a unique contribution to knowledge in four areas: (i) it will outline the development and implementation of a replicable teacher-training programme based on SDT, where strategies and techniques to support of three BPN are jointly designed and implemented in the teachers’ PE classes and which promote free-time PA; (ii) it will test the rarely proven effectiveness of SDT-based strategies delivered by PE teachers to promote Primary Education students’ participation in PA in their free time; (iii) through the different strategies, work among peers and the participation of families in physical-sports activities in their leisure time will be encouraged, thus trying to involve significant agents in the students’ learning process; and (iv) it will evaluate the long-term effectiveness of the intervention to promote PA behaviour through the four- and six-month post-intervention follow-up of behavioural and theory-based outcomes.

Despite the proven importance of PE for the development and learning of students in the practice of PA in their leisure time, so far, interventions to promote this through PE are scarce [8]. For example, a previous study [36] demonstrated the effectiveness of a PE-delivered intervention aimed at promoting PA participation in primary schools. However, the intervention program of that study included two weekly hours of PE, with a methodology of support for autonomy. In other words, strategies were used during PE classes to encourage students’ PA in their free time. In this sense, the present study will try to elaborate and put into practice strategies adapted to PE classes to promote out-of-school PA without modifying the programme initially established by the teachers. The study by Moreno-Murcia and Sánchez-Latorre [36] trains teachers to learn autonomy support techniques taught by another teacher. However, research has shown that student motivation can be strengthened if teachers support the three BPNs (i.e., autonomy, competence, and relatedness). On the other hand, motivation can be weakened and lead to maladaptive outcomes if teachers use thwarting techniques [59]. In our case, we propose a more complete training, addressing autonomy, competence and relatedness, and taught by a research team of university teachers whose line of research is linked to the SDT theoretical framework.

Beyond examining the expected outcomes of the intervention based on previous research, that is, increasing students’ motivation for PA and their leisure-time PA levels, this project’s findings can be used more holistically. For example, the programme is expected to generate students’ future habits of autonomous PA, improve their perception of competence in different PA, and motivate families to practice PA. Thus, considering the importance of promoting in- and out-of-school PA, we recommend developing SDT-based strategies in PE classes to promote an active life. In this sense, it is also worth highlighting the methodological benefits that this intervention will have in the school context. Firstly, with respect to the PE teachers who will receive a training programme in motivational theories (i.e., SDT), which will allow them to change or reinforce their approach to teaching classes in order to make their students more autonomous, competent and to improve social bonds both inside and outside of the school context. Furthermore, this training programme will also be important to encourage autonomous learning on the part of teachers so that they are in a continuum of renewal that allows them to adapt their classes to new social demands. Secondly, all the didactic materials and resources in the intervention programme will remain in the school for future years. This means that not only students in 4th- and 5th-grade of primary education will benefit from these strategies, but that students in other academic grades will have the possibility of taking advantage of these resources to improve their motivation towards PA in their leisure time, their engagement, their intention to be physically active and thus establish the bases so that this age group can increase their levels of PA.

## 4. Conclusions

It is expected that these school-based interventions aimed to promote the PA practice in the leisure time of students’ primary school will have positive consequences on them, such as increasing the levels of intrinsic motivation, increasing engagement during PE classes, reducing controlled motivation and demotivation towards PE and towards the practice of PA, and, consequently, there will be a positive predisposition towards the intention to carry out PA and increase the levels of PA practice of the students.

Regarding the first limitation that we can find in the development of the intervention, the need to have a supervisor in order to control the compliance and development of the strategies developed by the teachers may be a factor that may hinder the replication of similar interventions. Therefore, researchers who wish to carry out a similar intervention are urged to take into account the need or not to include an agent who can supervise the intervention to be carried out.

On the other hand, another of the limitations that can be found is due to the possible curricular modifications that the intervention may undergo as a result of the continuous legislative modifications in the educational context. Furthermore, it is important to highlight that each school has a different educational reality (i.e., the different organisation of the centre, practice spaces, quantity of PE teachers and students), and therefore, the strategies to be developed must be adapted and can be developed differently in the schools of the intervention programme.

Finally, taking into account the educational reality of PE teachers, it is important to emphasise that the teachers in this research have the theoretical and practical support of a research team. For this reason, it is encouraged to encourage PE teachers that if they do not have similar support behind them, they should carry out autonomous learning and training in motivational theories such as SDT in order to improve the teaching-learning process of their students.

## Figures and Tables

**Figure 1 children-10-00504-f001:**
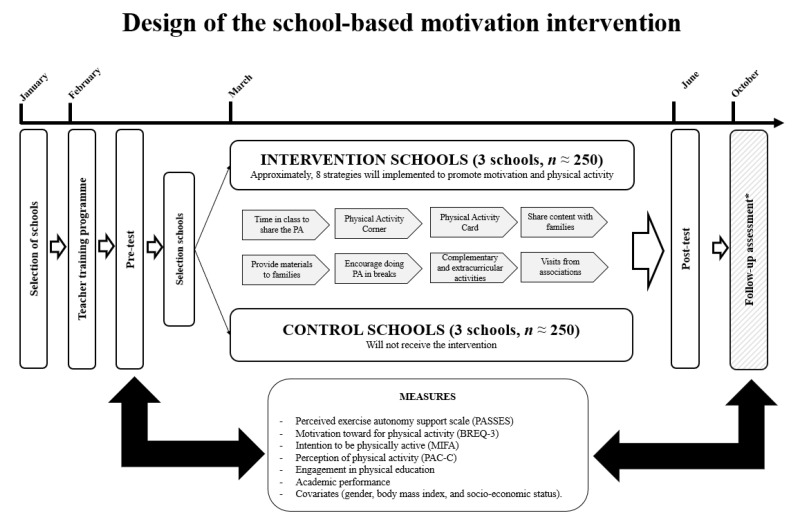
Timeline for data collection in months and schematic overview of study design.

**Figure 2 children-10-00504-f002:**
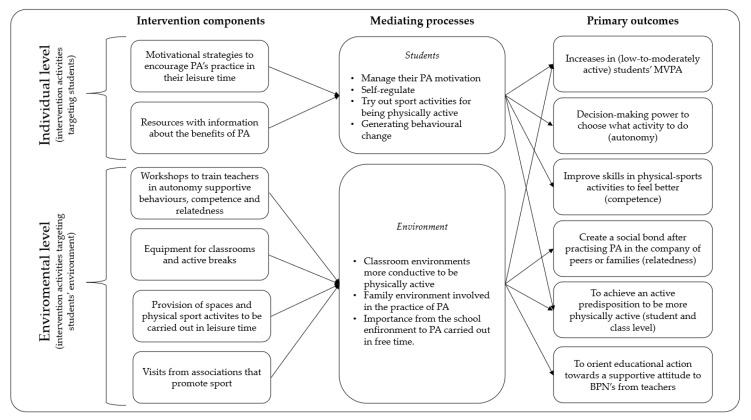
Streamlined coherent model linking intervention components to hypothesised mediating processes and primary outcomes.

## Data Availability

The datasets used and/or analysed during the current study are available from the corresponding author upon reasonable request.

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
