# Peer review of "Intervention Programme Based on Self-Determination Theory to Promote Extracurricular Physical Activity through Physical Education in Primary School: A Study Protocol"

_children, 2023, doi:10.3390/children10030504_

Round 1

Reviewer 1 Report

Reviewer’s comments on Intervention Programme based on Self-Determination Theory 2 to Promote Extracurricular Physical Activity through Physical 3 Education in Primary School: A Study Protocol

Reviewer’s comments on Intervention Programme based on Self-Determination Theory  to Promote Extracurricular Physical Activity through Physical 3 Education in Primary School: A Study Protocol

1.  General Comments for the Authors

The authors present interesting research on the motivation of schoolchildren for Physical Education. More specifically, if improvement as a result of the intervention exists; this could be a new step to enhance motivation for PE and to stimulate PA. 

In general, the idea is clear and the research method fits. Fairly enough, this could be informative for the readers However, if this paper is directed towards a discussion about a study protocol, the literature review should present the state of the art on study protocols.

2.  Specific Comments

Introduction

·      Line of reasoning: in line 43 it is stated that interventions on leisure time have been shown to be effective. There is no evidence in study 7 (Grao-Cruces et al., 2020) about altering motivation and its effect in or out of the classroom. It is merely a review study on PA.

·      It is logical to use the SDT as a foundation for motivation. It becomes less clear how these arguments are to use in this protocol. What is meant by the line “are needed to promote performance (and adherence) to PA? I would suggest elaborating a bit more on this assumption or narrowing it down.

·      The SDT is used to stimulate motivation. This is very much appreciated, but most of the literature to support this is either not directed toward PA ( e.g. Lazowski et al., 2016) or directed toward this target group (Gillison et al., 2017)). We would suggest rephrasing these sentences.

·      The evidence on the effect of SDT programs is overwhelming. A glance through these studies did reveal an enormous mixture of programs. Not all of them are directed toward PE classes or involve the three components (Hagger et al., 2005). Which is well commented on (lines 105-115). We would suggest explaining in a bit more detail why the use of PE teachers and PE lessons is that important or beneficial.

·      Overall, there is sufficient support for the SDT approach. It is less clear which teaching style and characteristics of the program are effective. Is there a minimum requirement for the duration or suggested teaching style? This can be presented in the introduction or the discussion.

Methodological

·      Concerning the chosen groups, it is not clear why this age group was chosen.

·      Is there ethical approval from a committee?

·      In figure 1 all kinds of elements are mentioned, but not addressed: f.i. Physical Activity corner. Furthermore, the PE lessons aren’t mentioned. Seems that there is no specific PE lesson.

·      The intervention program is based on choice and is based on autonomy. We greatly appreciate this personalized approach

·      Although there could be reasons to ask for background variables. This should be addressed in the methodology.

·      Some of the instrument present either reliability or validity issues (f.i. MIfa). We suggest addressing this at forehand and mentioning how to cope with this

·      Based on the different strategies the teachers use, it is unclear if all children get a different program

·      Some information should be given about the interpretation of effect sizes (GLM/ANOVA repeated measures)

·      The intervention seems very creative and inspiring. However, some attention should be directed toward the autonomy-supporting behavior of the teacher.

·      No information is given about the pre-assumptions of GLM

Discussion

·      With respect to the concise discussion, it’s recommended to add some points

·      Why is there no discussion about relatedness or competence stimulation?

·      As there is a process to develop the intervention, it seems logical to evaluate this process

·      What are the methodological merits and benefits of this study?

·      There is a holiday break in the study; this might affect the results.

·      What are the relationship/working mechanism between PA in leisure time and academic performance

Small remarks

·      There seems to be a mixed-use of synonyms describing the target group: children, youngsters, students; advice to use one word

Line 41: “encourage specific behaviors”; please be more concise: what behaviors are meant here

Line 27: “ to increase their students' adaptive outcomes.”; please define or elaborate

Line 210: “using PA as a nexus of the union”; please be more specific

Line 322: “proven importance”; explicit reference is failing

Author Response

February, 16th 2023

  1. General Comments for the Authors

The authors present interesting research on the motivation of schoolchildren for Physical Education. More specifically, if improvement as a result of the intervention exists; this could be a new step to enhance motivation for PE and to stimulate PA.

In general, the idea is clear and the research method fits. Fairly enough, this could be informative for the readers However, if this paper is directed towards a discussion about a study protocol, the literature review should present the state of the art on study protocols.

Many thanks to the reviewer for his or her important input. The state of the art on previous intervention programmes is presented in the introduction section. However, we may add further information if the reviewer considers the information provided to be insufficient.

  1. Specific Comments

Introduction

  • Line of reasoning: in line 43 it is stated that interventions on leisure time have been shown to be effective. There is no evidence in study 7 (Grao-Cruces et al., 2020) about altering motivation and its effect in or out of the classroom. It is merely a review study on PA.

Thanks to reviewer for the his or her suggestion. The sentence has been restructured to give more congruence to what we wanted to explain on the basis of the article by these authors.

Page 2, lines 46 – 48: “It has been shown that there is a need to develop strategies to encourage PA practice in students as PE levels during school hours are not enough [7].”

  • It is logical to use the SDT as a foundation for motivation. It becomes less clear how these arguments are to use in this protocol. What is meant by the line “are needed to promote performance (and adherence) to PA? I would suggest elaborating a bit more on this assumption or narrowing it down.

Thanks to the reviewer for his input. In accordance with your suggestion, this sentence has been clarified

Page 2, lines 55 - 57: “are needed to promote motivation and adherence toward to PA and alleviate dropout from PA over time [7]”

  • The SDT is used to stimulate motivation. This is very much appreciated, but most of the literature to support this is either not directed toward PA ( e.g. Lazowski et al., 2016) or directed toward this target group (Gillison et al., 2017)). We would suggest rephrasing these sentences.

We thank the reviewer for his or her important input. In accordance with his comment about the article by Lazowski et al. (2016), the sentence has been restructured in accordance with your request (P: 2; L: 60-61).

Page 2, lines 59 - 60: “Motivation has emerged as a crucial antecedent of students' development and learning in school context [10]”

With respect to the reference by Gillison et al. (2017), we did not intend to highlight the importance of developing interventions in the target group, but in the school environment in general. With regard to Primary Education, a brief description of the studies that are in line with the present study is set out in the following lines.

  • The evidence on the effect of SDT programs is overwhelming. A glance through these studies did reveal an enormous mixture of programs. Not all of them are directed toward PE classes or involve the three components (Hagger et al., 2005). Which is well commented on (lines 105-115). We would suggest explaining in a bit more detail why the use of PE teachers and PE lessons is that important or beneficial.

Thanks to reviewer for his/her indication. We have decided to include an explanatory paragraph about the importance of teachers in intervention programmes.

Page 3, lines 105-111: “Based on previous studies, one of the main limitations of school-based interventions is their low sustainability once the research team stops implementing the intervention. Although there are several reasons that may cause this abandonment of implementation, lack of teacher training is one of the main reasons. Therefore, previous literature suggests that training and involving teachers in the intervention programme could positively contribute to the sustainability of the school-based intervention [24].”

      Overall, there is sufficient support for the SDT approach. It is less clear which teaching style and characteristics of the program are effective. Is there a minimum requirement for the duration or suggested teaching style? This can be presented in the introduction or the discussion.

Thank you to reviewer for his/her indication. Firstly, with regard to the minimum duration required for a style intervention programme, we have included a sentence clarifying that there is no established duration and that in our study we have used other studies of interventions in Primary Education as a reference for estimating this duration.

Page 6, lines 213-215: “Given that there is no set duration as the benchmark for an intervention programme to be effective, it was decided to set a duration that was in line with previous programmes carried out in Primary Education [32,36–38]”.

To answer the second question about teaching style, it should be noted that there is a paragraph in the introduction which indicates the importance of teachers acquiring an interpersonal style supportive of basic psychological needs (P: 2-3; L: 94-104). Furthermore, this idea is reinforced in the discussion where the need for teachers to support autonomy, competence and social relationships in order to achieve adaptive student outcomes is again addressed (P: 9; L: 381 – 387).

Methodological

  • Concerning the chosen groups, it is not clear why this age group was chosen.

Thanks to the reviewer for his/her contribution. A sentence giving importance to the characteristics of the chosen group has been established in 2.1.3. Students' characteristics.

Page 5, lines 198-201: “The selection of these ages is based on the fact there are studies confirm that during the adolescence stage is when the number of sport dropout behavior occurs and there are higher rates of inactivity [34,35]. Hence the importance of generating good habits from an early age.”

  • Is there ethical approval from a committee?

Thank you again for your question. Yes, the approval by the ethics committee can be found on page 9, line 397 to 398. However, it has been decided to include it in the layout of the article for greater specificity (P: 4; L: 156-158).

Page 4, lines 156-158: “The study was conducted following the Declaration of Helsinki and was approved by the Ethics Committee of the University of Extremadura (120/2018).”

  • In figure 1 all kinds of elements are mentioned, but not addressed: f.i. Physical Activity corner. Furthermore, the PE lessons aren’t mentioned. Seems that there is no specific PE lesson.

Thanks to reviewer for his/her feedback. On the subject of strategies, Figure 1 sets out some examples of strategies that might emerge in training sessions with teachers because they are strategies that are in line with the development of the three basic psychological needs. These strategies, when it designed and implemented, will be developed in more detail in an intervention study. Furthermore, it should be noted that these strategies are not intended to alter the development of PE classes, but rather to be tools that allow for the promotion of the practice of PA in the students' free time. In other words, specific PE classes are not carried out in order to achieve the proposed objectives.

  • The intervention program is based on choice and is based on autonomy. We greatly appreciate this personalized approach

Thanks to reviewer for this appreciation of the intervention programme.

  • Although there could be reasons to ask for background variables. This should be addressed in the methodology.

Thanks to the reviewer for his or her input. The study variables are described in subsection 2.3 of the study methodology. However, if the reviewer does not consider the description to be incomplete, please let us know and we may be able to expand it.

  • Some of the instrument present either reliability or validity issues (f.i. MIfa). We suggest addressing this at forehand and mentioning how to cope with this

Thanks to reviewer for his/her concern about the reliability and validity of the instruments. Also, in line with your comment on the MIFA questionnaire, previous studies have been found to show adequate reliability of the instrument: (Cronbach's Alpha >.70; Aguirre et al., 2018; Franco Álvarez et al., 2016; García-Cerberino et al., 2021; Grao-Cruces et al., 2017).

Franco Álvarez, E., Coterón López, J., & Pérez-Tejero, J. (2016). Intención de ser físicamente activos entre estudiantes de EF: diferencias según la obligatoriedad de la enseñanza. Revista Española de Educación Física y Deportes, 414, 39–51. https://doi.org/https://doi.org/10.55166/reefd.v0i414.479

García-Cerberino, J. M., Gamero, M. G., Feu, S., & Ibáñez, S. J. (2021). La percepción de la competencia en fútbol como indicador de la intencionalidad de los estudiantes de ser físicamente activos. E-Balonmano - Revista de Ciencias Del Deporte, 17(1), 1–12.

Grao-Cruces, A., Fernández-Martínez, A., Teva-Villén, M. R., & Nuviala, A. (2017). Autoconcepto físico e intencionalidad para ser físicamente activo en los participantes del programa Escuelas Deportivas. Journal of Sport and Health Research, 9(1), 15–26.

For the rest of the variables, we followed the same procedure, taking into account previous studies.

  • Perceived Need Support in PA (Ferriz et al., 2020; González-Cutre et al., 2020)

Ferriz, R., González-Cutre, D., & Balaguer-Giménez, J. (2020). Agentes sociales de la comunidad educativa, satisfacción de novedad y actividad física (Agents of the educational community, novelty satisfaction, and physical activity). Cultura, Ciencia y Deporte, 15(46). https://doi.org/10.12800/ccd.v15i46.1602

González-Cutre, D., Jiménez-Loaisa, A., Alcaraz-Ibáñez, M., Romero-Elías, M., Santos, I., & Beltrán-Carrillo, V. J. (2020). Motivation and physical activity levels in bariatric patients involved in a self-determination theory-based physical activity program. Psychology of Sport and Exercise, 51, 101795. https://doi.org/10.1016/j.psychsport.2020.101795

  • Motivation for Physical Activity (Durán-Vinagre et al., 2023; Romero-Parra et al., 2022).

Durán-Vinagre, M. Á., Ibáñez, S. J., Feu, S., & Sánchez-Herrera, S. (2023). Analysis of the motivational processes involved in university physical activity. Frontiers in Psychology, 13. https://doi.org/10.3389/fpsyg.2022.1080162

Romero-Parra, N., Solera-Alfonso, A., Bores-García, D., & Delfa-de-la-Morena, J. M. (2022). Sex and educational level differences in physical activity and motivations to exercise among Spanish children and adolescents. European Journal of Pediatrics, 182(2), 533–542. https://doi.org/10.1007/s00431-022-04742-y

  • Perception of Physical Activity Levels (Penagini et al., 2022; Vandoni et al., 2022).

Penagini, F., Calcaterra, V., Dilillo, D., Vandoni, M., Gianolio, L., Gatti, A., Rendo, G., Giuriato, M., Cococcioni, L., De Silvestri, A., & Zuccotti, G. (2022). Self-Perceived Physical Level and Fitness Performance in Children and Adolescents with Inflammatory Bowel Disease. Children, 9(9), 1399. https://doi.org/10.3390/children9091399

Vandoni, M., Carnevale Pellino, V., Gatti, A., Lucini, D., Mannarino, S., Larizza, C., Rossi, V., Tranfaglia, V., Pirazzi, A., Biino, V., Zuccotti, G., & Calcaterra, V. (2022). Effects of an Online Supervised Exercise Training in Children with Obesity during the COVID-19 Pandemic. International Journal of Environmental Research and Public Health, 19(15), 9421. https://doi.org/10.3390/ijerph19159421

  • Engagement in Physical Education (Howle et al., 2015; Verma et al., 2019).

Howle, T. C., Dimmock, J. A., Whipp, P. R., & Jackson, B. (2015). The Self-Presentation Motives for Physical Activity Questionnaire: Instrument Development and Preliminary Construct Validity Evidence. Journal of Sport and Exercise Psychology, 37(3), 225–243. https://doi.org/10.1123/jsep.2014-0134

Verma, N., Eklund, R. C., Arthur, C. A., Howle, T. C., & Gibson, A.-M. (2019). Transformational Teaching, Self-Presentation Motives, and Identity in Adolescent Female Physical Education. Journal of Sport and Exercise Psychology, 41(1), 1–9. https://doi.org/10.1123/jsep.2017-0299

  • Based on the different strategies the teachers use, it is unclear if all children get a different program

Thanks to reviewer for his/her appreciation. Based on your input, it has been decided to clarify this sentence.   

Page 6, lines 233-235: “Each experimental school will carry out the same strategies. However, each teacher will be able to apply them as they see fit, adapting them to their own educational reality, under the supervision of a member of the research team”.

  • Some information should be given about the interpretation of effect sizes (GLM/ANOVA repeated measures)

We thank the reviewer for his important contribution. In line with his comment, the interpretation of effect sizes has been added.

Pages 8 – 9, lines 332 – 343: “For continuous variables, Bonferroni-corrected multiple paired t-tests will be calculated to determine intra-group (i.e., between experimental and control group) and inter-group (i.e., between pre-test and post-test) differences. For categorical variables, a chi-square test will be performed. While Cramer's V will be used to describe the degree of association between categorical variables and experimental and control schools, McNemar's test will be used to analyse baseline and post-intervention differences in categorical variables in experimental and control schools, respectively. Effect sizes will be assessed using Partial Eta Squared (ηp2) and Cramer's V values for continuous and categorical variables, respectively. Effect sizes will be considered small, moderate or large when ηp2 are greater than 0.01, 0.06 and 0.14, respectively, and when Cramer's V are greater than 0.10, 0.30 and 0.50, respectively.”

  • The intervention seems very creative and inspiring. However, some attention should be directed toward the autonomy-supporting behavior of the teacher.

Thanks to reviewer 1 for this appreciation. There are two key elements that serve to guide the teacher's autonomy-supportive behaviour: 1) The training programme through which they will receive SDT-based learning and the importance of such support; 2) The figure of the supervisor within the research group in order to be able to conduct needs-supportive behaviours.

  • No information is given about the pre-assumptions of GLM

Thank you for your input. This information has been added to the statistical analysis section.

Page 8, lines 328 – 330: “Secondly, the principles of independence, normality, linearity and variance of the study variables will be tested for general model analyses.”

Discussion

  • With respect to the concise discussion, it’s recommended to add some points

Thanks to reviewer 1 for this comment. On this basis, different sections have been included in the discussion that have allowed for more information in this section. All changes are visible in the document.

  • Why is there no discussion about relatedness or competence stimulation?

Thanks to reviewer for the question. The discussion refers to the differences between our study and those previously conducted by other authors. These differences include the need for a training programme based on autonomy, competence and relatedness, and the importance of stimulating all three to encourage the development of the three BPNs in students (P: 9; L: 378 - 387).

  • As there is a process to develop the intervention, it seems logical to evaluate this process

Thanks to reviewer for the suggestion. A clarifying paragraph has been included regarding the evaluation of the process and development of the intervention programme.

Page 8, lines 320 - 323: “In addition, to evaluate the process and development of the school-based intervention, semi-structured interviews will be carried out with teachers, students and families in order to obtain complementary information to that obtained from the different questionnaires and scales mentioned above.”

  • What are the methodological merits and benefits of this study?

Thanks to reviewer for this question. Regarding the question presented, we would like to point out that the methodological benefits can be divided on the basis of two main agents:

Firstly, teachers who are going to receive a training day (i.e., SDT) that is going to allow them to receive learning that can change or reinforce their methodological perspective, based on the support of the three BPNs. Secondly, not only the students who will receive the different strategies, but also the students of other grades will be able to benefit from all the materials and didactic resources developed since they will remain in the school for use in the following school years.

However, it has been decided to include a paragraph explaining the above-mentioned benefits in detail.

Page 9-10, lines 387-401: “In this sense, it is also worth highlighting the methodological benefits that this intervention will have in the school context. Firstly, with respect to the PE teachers who will receive training programme in motivational theories (i.e., SDT) that will allow them to change or reinforce their approach to teaching classes in order to make their students more autonomous, competent and to improve social bonds inside and out-of-school con-text. Furthermore, this training programme will also be important to encourage autonomous learning on the part of teachers so that they are in a continuum of renewal that al-lows them to adapt their classes to new social demands. Secondly, all the didactic mate-rials and resources in the intervention programme will remain in the school for future years. This means that not only students in 4th- and 5th-grade of Primary Education will benefit from these strategies, but that students in other academic grades will have the possibility of taking advantage of these resources to improve their motivation towards PA in their leisure time, their engagement, their intention to be physically active and thus establish the bases so that this age group can increase their levels of PA.”.

  • There is a holiday break in the study; this might affect the results.

Thank you for your appreciation. We understand the concern of reviewer regarding this holiday break. The aim of the intervention programme being developed before the holiday break is to provide students with sufficient tools and resources for them to carry out PA independently in their free time, being extrapolated to this holiday period. The follow-up measure in the next school year will allow us to analyse the influence of this holiday period on the practice of PA by the students and analyse the long-term effects.

  • What are the relationship/working mechanism between PA in leisure time and academic performance

Thanks to reviewer for the appreciation. Authors have included more information to describe the positive effect between PA in leisure time and academic performance.

Page 2, lines 92 - 93: “In this sense, PA and fitness are related to different types of changes in the brain relevant for cognitive function and learning [21–24].”.

Small remarks

  • There seems to be a mixed-use of synonyms describing the target group: children, youngsters, students; advice to use one word

Thank you to the reviewer for his/her indication. The decision has been taken to unify the whole document with “student”.

Line 41: “encourage specific behaviors”; please be more concise: what behaviors are meant here

Thanks to reviewer again. A parenthesis has been included exemplifying some of these specific behaviours.

Page 1, lines 43-44: “(e.g., cognitive and academic development, development at behavioural, cellular, functional and morphological levels).”

Line 27: “ to increase their students' adaptive outcomes.”; please define or elaborate

Thanks to reviewer for this appreciation. Since the abstract cannot refer to everything that is intended to be explained in the protocol, in order not to be redundant it was decided to refer to these adaptive student outcomes since they are explained concretely in the discussion of this study (P: 9; L: 381-387).

Line 210: “using PA as a nexus of the union”; please be more specific

Thanks to reviewer for his/her suggestion. It has been decided to include an explanatory sentence on the importance of PA as a tool to connect people socially with the environment around them.

Page 6, lines 230 - 231: “between people who intend to practice physical-sports activities with other people for greater enjoyment.”

Line 322: “proven importance”; explicit reference is failing

Thanks to reviewer again. The reference to the study by Kelso et al. (2020) has been included to magnify this sentence (P: 8; L: 366).

Kelso, A.; Linder, S.; Reimers, A.K.; Klug, S.J.; Alesi, M.; Scifo, L.; Borrego, C.C.; Monteiro, D.; Demetriou, Y. Effects of School-Based Interventions on Motivation towards Physical Activity in Children and Adolescents: A Systematic Review and Meta-Analysis. Psychol. Sport Exerc. 2020, 51, 101770, doi:10.1016/j.psychsport.2020.101770.

Reviewer 2 Report

I have carefully read the manuscript and my opinion is that the manuscript has a merit to be published in your reputable journal with some minor corrections. The manuscript is original, informative and readable. The authors tried to present a protocol intervention in Physical Education (PE) based on Self-Determination Theory (SDT) to enhance children's motivation towards performing PA and increasing their PA levels in their leisure time. Although I did not review too many manuscripts that were study protocols, I am not too experienced but I will try to give some recommendations to improve this manuscript. At the first place, the abstract is well prepared but I would prefer not to use future tense if the tasks is completed. Otherwise, I do not understand why the authors are trying to present something that is not completed yet?! The introduction part is also well written, the authors reviewed well the physical activity and self-determination theory as well as previous intervention studies based on SDT in the school context, and finalized this section by highlights from previous studies. However, I do not understand the second part of the introduction section as it is described in the future tense again. If the protocol study represent the promotion of the future protocol, in that case, I have no objections at all. Otherwise, it is a kind of confusing me. The method sections is also in the future tense (present tense would be more suitable from my point of view), I do recommend to present the protocol as it is already prepared, although it is not already used. Furthermore, I have no amendments on discussion part but I would recommend to the authors to prepare the conclusion part and consider to structure it in the following order: the main conclusions, the potential limitations of the protocol (more precisely) as well as some recommendations for the further potential users of the protocol as well as for the future investigations. It is also very important to highlight that the quality of the figures and tables is satisfactory.

Author Response

February, 16th 2023

I have carefully read the manuscript and my opinion is that the manuscript has a merit to be published in your reputable journal with some minor corrections. The manuscript is original, informative and readable. The authors tried to present a protocol intervention in Physical Education (PE) based on Self-Determination Theory (SDT) to enhance children's motivation towards performing PA and increasing their PA levels in their leisure time. Although I did not review too many manuscripts that were study protocols, I am not too experienced but I will try to give some recommendations to improve this manuscript. At the first place, the abstract is well prepared but I would prefer not to use future tense if the tasks is completed. Otherwise, I do not understand why the authors are trying to present something that is not completed yet?! The introduction part is also well written, the authors reviewed well the physical activity and self-determination theory as well as previous intervention studies based on SDT in the school context, and finalized this section by highlights from previous studies. However, I do not understand the second part of the introduction section as it is described in the future tense again. If the protocol study represent the promotion of the future protocol, in that case, I have no objections at all. Otherwise, it is a kind of confusing me. The method sections is also in the future tense (present tense would be more suitable from my point of view), I do recommend to present the protocol as it is already prepared, although it is not already used. Furthermore, I have no amendments on discussion part but I would recommend to the authors to prepare the conclusion part and consider to structure it in the following order: the main conclusions, the potential limitations of the protocol (more precisely) as well as some recommendations for the further potential users of the protocol as well as for the future investigations. It is also very important to highlight that the quality of the figures and tables is satisfactory.

First assessment – Verb tense

Dear reviewer, thank you for your concern about the verb tense used in this protocol. We would like to tell you that those indicators referring to past tenses have already been modified because the protocol has to be written in the future tense. You can look at the protocol of another educational intervention carried out by some of the authors of the present study to see that the most frequently used verb tense is the future tense and, occasionally, reference is made to present simple.

Demetriou, Y., & Bachner, J. (2019). A school-based intervention based on self-determination theory to promote girls’ physical activity: study protocol of the CReActivity cluster randomised controlled trial. BMC Public Health, 19(1), 519. https://doi.org/10.1186/s12889-019-6817-y

Sánchez-Miguel, P. A., Vaquero-Solís, M., Sánchez-Oliva, D., Pulido, J. J., López-Gajardo, M. A., & Tapia-Serrano, M. A. (2020). Promoting Healthy Lifestyle through Basic Psychological Needs in Inactive Adolescents: A Protocol Study from Self-Determination Approach. Sustainability, 12(15), 5893. https://doi.org/10.3390/su12155893

Second assessment – Conclusion section

Thanks to the reviewer for the suggestion to include a concluding section to end the article. It has therefore been decided to include this information on page 10, from line 402 to line 427. This section has been divided into two sections. Firstly, a paragraph on the main results expected to be found after the intervention:

Page 10, lines 403 - 408: “It is expected that these school-based interventions aimed to promote the PA’ practice in the leisure time of students’ Primary School will have positive consequences on them, such as increasing the levels of intrinsic motivation, increasing engagement during PE classes, reducing controlled motivation and demotivation towards PE and towards the practice of PA, and, consequently, there will be a positive predisposition towards the intention to carry out PA and increase the levels of PA practice of the students.”

Secondly, a series of possible limitations that the study may have during its development is established, as well as the recommendations that serve to alleviate these limitations:

Page 10, lines 409 - 427: “Regarding the first limitation that we can find in the development of the intervention, the need to have a supervisor in order to control the compliance and development of the strategies developed by the teachers may be a factor that may hinder the replication of similar interventions. Therefore, researchers who wish to carry out a similar intervention are urged to take into account the need or not to include an agent who can supervise the intervention to be carried out.

On the other hand, another of the limitations that can be found is due to the possible curricular modifications that the intervention may undergo as a result of the continuous legislative modifications in the educational context. Furthermore, it is important to high-light that each school has a different educational reality (i.e., the different organisation of the centre, practice spaces, number of PE teachers and number of students) and therefore, the strategies to be developed must be adapted and can be developed differently in the schools of the intervention programme.

Finally, taking into account the educational reality of PE teachers, it is important to highlight that the teachers in this research have the theoretical and practical support of a research team. For this reason, it is encouraged to encourage PE teachers that if they do not have similar support behind them, they should carry out autonomous learning and training in motivational theories such as SDT in order to improve the teaching-learning process of their students.”

Reviewer 3 Report

Dear friends, after reading and analyzing the study. The topic addressed is relevant and the study has been adequately detailed. My minor considerations are as follows:

1. I suggest informing the level of significance that will be adopted in all analyses;

2. Was a sample calculation performed? It would be important to show readers the exact number of participants needed to achieve power in the assessments;

3. Figure 1 must be crafted! Images are too dark. This does not facilitate the understanding of the activities.

Author Response

February, 26th 2023

Dear Donna Ding,

Please, find attached a revision of our manuscript entitled “Intervention Programme based on Self-Determination Theory to Promote Extracurricular Physical Activity through Physical Education in Primary School: A Study Protocol”. We would like to thank reviewers for their constructive comments and suggestions on the manuscript. We have considered all suggestions and have included them in the revised manuscript using red colour. We believe that our manuscript is stronger and more coherent as a result of these modifications. Below, we explain, one by one, the modifications we have made as a result of the reviewer’s comments. In addition, the language has been reviewed by a native translator, whose certificate is attached. We hope that the manuscript may be of interest to Children readers.

Best regards,

Rubén Llanos Muñoz

University of Extremadura

Teacher Training College

 “Intervention Programme based on Self-Determination Theory to Promote Extracurricular Physical Activity through Physical Education in Primary School: A Study Protocol”

Reviewer 3

Dear friends, after reading and analyzing the study. The topic addressed is relevant and the study has been adequately detailed. My minor considerations are as follows:

  1. I suggest informing the level of significance that will be adopted in all analyses

Thanks to reviewer for his/her suggestion. Corresponding information has been included.

                Page 9, Line 353: “Significance will set at p < 0.05 for all analysis.”

  1. Was a sample calculation performed? It would be important to show readers the exact number of participants needed to achieve power in the assessments;

Thanks to reviewer for his/her question. Corresponding information has been added in the design and participants section.

Page 4, Line 149-153: “The sample size will estimate through the following formula: n = (Z)2(p (1 - p) e2), where n is the sample size, Z = 1.96 (95% confidence interval), p = number of 4th- and 5th-primary school students in the city where the study will conduct (± 3350 students), and e = margin of error (3%). The minimum sample size (considering a 10% non-response) will be 345 students.”

  1. Figure 1 must be crafted! Images are too dark. This does not facilitate the understanding of the activities.

Thanks to reviewer again. Figure 1 has already been modified for better understanding and readability. 

Round 2

Reviewer 1 Report

Thanks for the alterations and sufficiently answering the recommendations. Looking forward to read the article in the published form

Author Response

The authors would like to thank you for your kind contributions to our manuscript.